# Treatment of hypertension by increasing impaired endothelial TRPV4-KCa2.3 interaction

Dongxu He[1,2,†], Qiongxi Pan[1,2,†], Zhen Chen[1,2,†], Chunyuan Sun[1,2,†], Peng Zhang[1,2], Aiqin Mao[1,2], Yaodan Zhu[1,2], Hongjuan Li[1,2], Chunxiao Lu[1,2], Mingxu Xie[1,2], Yin Zhou[1,2], Daoming Shen[1,2], Chunlei Tang[1,2], Zhenyu Yang[3], Jian Jin[1,2], Xiaoqiang Yao[4], Bernd Nilius[5] & Xin Ma[1,2,*]

## Abstract

The currently available antihypertensive agents have undesirable adverse effects due to systemically altering target activity including receptors, channels, and enzymes. These effects, such as loss of potassium ions induced by diuretics, bronchospasm by beta-blockers, constipation by $Ca^{2+}$ channel blockers, and dry cough by ACEI, lead to non-compliance with therapies (Moser, 1990). Here, based on new hypertension mechanisms, we explored a new anti-hypertensive approach. We report that transient receptor potential vanilloid 4 (TRPV4) interacts with $Ca^{2+}$-activated potassium channel 3 (KCa2.3) in endothelial cells (ECs) from small resistance arteries of normotensive humans, while ECs from hypertensive patients show a reduced interaction between TRPV4 and KCa2.3. Murine hypertension models, induced by high-salt diet, N(G)-nitro-L-arginine intake, or angiotensin II delivery, showed decreased TRPV4-KCa2.3 interaction in ECs. Perturbation of the TRPV4-KCa2.3 interaction in mouse ECs by overexpressing full-length KCa2.3 or defective KCa2.3 had hypotensive or hypertensive effects, respectively. Next, we developed a small-molecule drug, JNc-440, which showed affinity for both TRPV4 and KCa2.3. JNc-440 significantly strengthened the TRPV4-KCa2.3 interaction in ECs, enhanced vasodilation, and exerted antihypertensive effects in mice. Importantly, JNc-440 specifically targeted the impaired TRPV4-KCa2.3 interaction in ECs but did not systemically activate TRPV4 and KCa2.3. Together, our data highlight the importance of impaired endothelial TRPV4-KCa2.3 coupling in the progression of hypertension and suggest a novel approach for antihypertensive drug development.

**Keywords** artery; endothelium; hypertension; KCa2.3; TRPV4
**Subject Categories** Cardiovascular System; Vascular Biology & Angiogenesis

## Introduction

Currently available antihypertensive drugs, which include diuretics, beta-blockers, angiotensin-converting enzyme inhibitors, angiotensin receptor blockers, and $Ca^{2+}$ channel blockers, exert effects on cardiac output or sympathetic neural activity, resulting in reduced blood pressure. In addition to these pathogenic mechanisms of disease, which are targeted by existing antihypertensive drugs, changes in the vascular resistance of small resistance arteries have been implicated in the development of hypertension (Moser, 1990; Panza *et al*, 1990; Wong *et al*, 2004). Notably, however, the critical role of endothelial cells (ECs) in small resistance arteries in the pathogenesis of hypertension has been largely ignored to date.

In hypertension, functional changes in small resistance arteries are frequently accompanied by abnormalities of the ECs in these vessels (Feletou & Vanhoutte, 2006a). While endothelium-dependent relaxation is selectively mediated by nitric oxide in the large conduit vessels, endothelium-derived hyperpolarizing factor (EDHF) is the predominant vasodilator in small resistance arteries (Brandes *et al*, 2000; Busse *et al*, 2002). An absence of EDHF increases peripheral resistance and blood pressure (Panza *et al*, 1990; Feletou *et al*, 2010); in turn, the endothelial dysfunction is reinforced, and then compliance and pressure reflectivity in the arteries are attenuated, which further increases blood pressure, thereby causing a vicious cycle (Laurent *et al*, 2006; Najjar *et al*, 2008). Therefore, targeting the function of small arteries via the EDHF signaling pathway tends to be an effective means of treating hypertension (Feletou & Vanhoutte, 2009).

Endothelial $K^+$ channels, including the small conductance and intermediate conductance $Ca^{2+}$-activated $K^+$ channels, contribute significantly to the EDHF response (Burnham *et al*, 2002; Bychkov *et al*, 2002; Garland *et al*, 2011; Sonkusare *et al*, 2012), and the presence of TRPV4 is essential for this response (Watanabe *et al*, 2003; Vriens *et al*, 2005; Earley *et al*, 2009; Bagher *et al*, 2012). We previously demonstrated that physical and functional interaction of TRPV4 with the $Ca^{2+}$-activated $K^+$ channel KCa2.3 is responsible for the EDHF response in ECs (Ma *et al*, 2013). This type of channel

1  School of Medicine, Jiangnan University, Wuxi, China
2  National Engineering Laboratory for Cereal Fermentation Technology, Jiangnan University, Wuxi, China
3  Heart Centre, Wuxi People's Hospital, Wuxi, China
4  School of Biomedical Sciences, The Chinese University of Hong Kong, Shatin, Hong Kong
5  Department Cell Mol Medicine Laboratory Ion Channel Research Campus Gasthuisberg, KU Leuven, Leuven, Belgium
   *Corresponding author. Tel: +86 510 85914599; E-mail: maxin@jiangnan.edu.cn
   †These authors contributed equally to this work

association allows $Ca^{2+}$ entry through TRPV4 channels to stimulate neighboring KCa2.3 channels in subcellular microdomains, evoking the EDHF response and contributing to the subsequent vascular relaxation and regulation of blood flow and blood pressure. Here, with the goal of effectively treating hypertension and cutting down on the accompanying adverse effects due to systemic activation, we explored the mechanism by which hypertension is caused by dysregulated TRPV4-KCa2.3 interaction; we also identified a potentially therapeutic small molecule like a pin targeting impaired endothelial TRPV4-KCa2.3 interaction (Fig 1A).

## Results and Discussion

### Impairment of the physical and functional interaction of TRPV4-KCa2.3 channels in hypertension

To examine the role of TRPV4-KCa2.3 interaction in hypertension, we assessed the interaction between these two proteins in ECs from human arterial segments using Förster resonance energy transfer (FRET) (Adebiyi et al, 2010, 2012) and found that the physical interaction between TRPV4-KCa2.3 was lower in cells from hypertensive patients than in those from normotensive individuals (Fig 1B, Appendix Fig S1A and B, and Appendix Table S1). Next, we treated mice with a high-salt diet, N(G)-nitro-L-arginine (L-NNA) intake, or angiotensin II (AngII) delivery to generate murine hypertension models, and the FRET results showed that TRPV4-KCa2.3 interaction was similarly reduced in the hypertensive and control mice (Fig 1C–E). The TRPV4-KCa2.3 interaction in mouse ECs was also confirmed by co-immunoprecipitation (co-IP) (Appendix Fig S2A), which also showed that the interaction was decreased in hypertensive mice (Appendix Fig S2B).

We previously showed that the function of KCa2.3 is dependent on TRPV4 and is responsible for the EDHF response in ECs (Ma et al, 2013). Here, we further explored whether destruction of physical TRPV4-KCa2.3 interaction abolished their functional interaction.

In mouse primary arterial ECs, the specific TRPV4 agonist GSK1016790A dose-dependently induced $K^+$ efflux (Fig 1F), indicating the functional interaction of TRPV4 with KCa2.3. Consistently, whole-cell patch clamp recordings from primarily cultured ECs showed that GSK1016790A induced a whole-cell current, which was inhibited by the TRPV4 inhibitor HC067047, but not by the KCa2.3 inhibitor apamin (Appendix Fig S3A). In contrast, the KCa2.3 activator CyPPA induced a whole-cell current, which was inhibited by the TRPV4 inhibitor HC067047 (Appendix Fig S3B), suggesting again the functional interaction of TRPV4-KCa2.3 and the dependence of KCa2.3 function on TRPV4. We then showed that the functional interaction of TRPV4 and KCa2.3 including $K^+$ efflux in ECs (Fig 1F) and the dilation in freshly isolated arterial segments induced by GSK1016790A was decreased in mice that had been fed a high-salt diet (Fig 1G). These data suggest that the interrupted physical TRPV4-KCa2.3 interaction consequently impaired the functional TRPV4-KCa2.3 interaction.

To clarify the involvement of EDHF in this TRPV4-KCa2.3 mechanism in hypertension, we showed in freshly isolated small arterial segments from mice on a high-salt diet, acetylcholine (ACh)-induced vasodilation was weakened, but sodium nitroprusside (SNP)-induced vasodilation was not significantly changed (Fig 1G). These results are consistent with previous reports (Barron et al, 2001) and suggest that a high-salt diet weakens EDHF-dependent, but not nitric oxide-dependent vasodilation. Consistently, by measuring membrane potentials via sharp microelectrodes inserted from the adventitial side in small arteries, we showed that the smooth muscle hyperpolarization, which is an indicator of EDHF release (Busse et al, 2002; Feletou & Vanhoutte, 2006b, 2009), was activated by GSK1016790A or ACh. An antagonist of KCa2.3 (apamin) or TRPV4 (HC067047) inhibited ~80% of the hyperpolarization induced by GSK1016790A or ACh (Appendix Fig S4). Thus, these data suggest that EDHF is the dominant component here and that TRPV4-KCa2.3 contributes to such EDHF signaling.

We further attempted to identify the domains/regions involved in TRPV4-KCa2.3 interaction using deletion assays and FRET in

**Figure 1. Impairment of the physical and functional interaction of TRPV4-KCa2.3 channels in hypertension.**

A  The hypothesis investigated in this study was that endothelial TRPV4-KCa2.3 interaction modulates blood pressure, and impairment of the interaction is related to hypertension. So, strengthening the interaction with a small-molecule drug, JNc-440, would show antihypertensive effects.

B  Immuno-FRET experiments in arterial sections from normotensive and hypertensive humans. Left: representative figures for FRET. White dotted line, autofluorescence of elastin underneath the endothelium; right: analysis of 21 normotensive and 15 hypertensive individuals ([#]$P < 0.0001$, unpaired t-test vs. control). Scale bar, 20 μm.

C–E  Immuno-FRET experiments in arterial sections from (C) mice on a high-salt diet (8% NaCl), (D) mice with L-NNA intake (0.5 g/l in the drinking water), or (E) mice with AngII delivery (infused at 500 ng/kg/min) ([#]$P < 0.0001$, unpaired t-test vs. control).

F  Fluorescent $K^+$ efflux results showing the TRPV4 agonist GSK1016790A (GSK)-induced $K^+$ efflux in primary cultured ECs from normal mice and mice on a high-salt diet (left). Six mice were tested for each treatment and analyzed (right). ECs from TRPV4-knockout (KO) mice were used as negative controls [[#]$P < 0.0001$, two-way ANOVA (GSK 30 nM, high salt vs. control; GSK 100 nM, high salt vs. control)].

G  Modulation of vasodilation by GSK1016790A (GSK), acetylcholine (ACh), and sodium nitroprusside (SNP). Left: vasodilation assays by wire myograph showing GSK1016790A (GSK)-induced dilation in freshly isolated arterial segments from normal mice and mice on a high-salt diet. Second from left: five mice were tested for each treatment and statistically analyzed ([#]$P < 0.0001$ vs. control, two-way ANOVA). Rightmost two panels: statistical data of vasodilation for ACh ($n = 3$/treatment) and SNP ($n = 3$/treatment) treatment in freshly isolated arterial segments from normal mice and mice on a high-salt diet (*$P < 0.0001$, [#]$P = 0.0004$ vs. control, two-way ANOVA).

H  FRET in HEK cells co-expressing CFP-tagged TRPV4 and YFP-tagged KCa2.3 showing the critical role of the AR2 and C-terminal 17-amino acid (17C) regions in the interaction. TRPV4 was deleted at the different AR and CaMBD regions, whereas KCa2.3 was deleted at the 17C, α1, α2, Loop, and CAMBD regions. NC, negative control, assessed in cells co-expressing CFP and YFP as separate molecules; PC, positive control, assessed in cells expressing YFP-CFP dimer ([#]$P < 0.0001$ vs. wild-type control (TRPV4-KCa2.3), one-way ANOVA).

I  Stochastic optical reconstruction images of the indicated constructs in overexpressing HEK cells (scale bars, 10 μm).

Data information: Data are given as the mean ± SD.

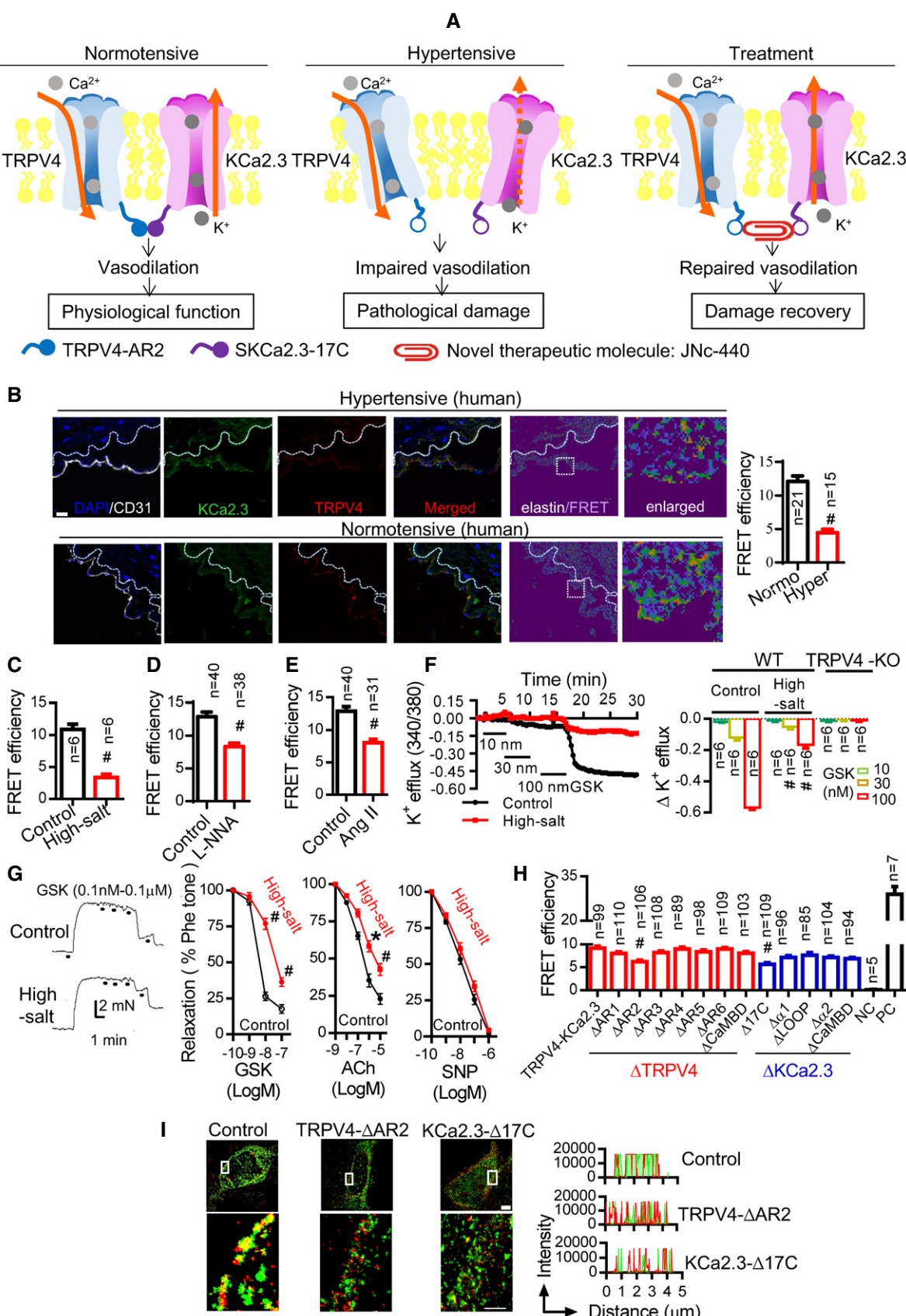

**Figure 1.**

HEK cells co-expressing CFP-tagged TRPV4 and YFP-tagged KCa2.3. The N-terminal ankyrin repeat domain (AR) of TRPV4 and the CaM-binding domain (CaMBD) of KCa2.3 are known to be critical for their function (Schumacher et al, 2001; Strotmann et al, 2003; Cifuentes et al, 2004; Stocker, 2004; Roncarati et al, 2005; Zhu, 2005; Phelps et al, 2010; Takahashi et al, 2014). Deletion of either the AR2 cytoplasmic region of TRPV4 (ΔAR2) or a 17-amino acid region in the C-terminal cytoplasmic region of KCa2.3 (Δ17C), but not of various other regions within these two channels, significantly decreased the FRET signal, indicating that these regions are responsible for the interaction (Fig 1H). Mutation of either protein had no evident effect on channel function or cellular localization (Appendix Figs S5 and S6). The critical role of the TRPV4-AR2 and the KCa2.3-17C regions in the interaction was confirmed by stochastic optical reconstruction microscopy (Fig 1I). Together, these results indicated that in the setting of hypertension, physical and functional interaction of TRPV4 and KCa2.3 in ECs is reduced, and the TRPV4-AR2 and KCa2.3-17C regions are critical for this interaction (Fig 1).

We then investigated the mechanism by which TRPV4 and KCa2.3 become disassociated in hypertension. Caveolin-1 is the major structural protein of caveolae, and acts as a scaffold for clusters of ion channels and signaling molecules within them. Within the caveolin-1 scaffold of EC caveolae, the $Ca^{2+}$-dependent effector KCa2.3 resides within the spatial boundaries of the $Ca^{2+}$-permeable TRPV4 channels. This subcellular arrangement may enhance the ability of KCa2.3 to sense $Ca^{2+}$ influx through TRPV4 channels (Loot et al, 2008; Saliez et al, 2008; Goedicke-Fritz et al, 2015). In the setting of hypertension, the caveolin-1 scaffold is disrupted (Mathew et al, 2004; Zhang et al, 2014b), and we hypothesized that this disruption is accompanied by physical separation of TRPV4 and KCa2.3. To test this, we first confirmed that caveolin-1 associated with TRPV4 and KCa2.3 and that TRPV4 and KCa2.3 associated with each other in the caveolae of murine ECs under normal conditions (Appendix Figs S2A and S7). Next, we treated mice with high-salt diet, L-NNA intake, or AngII delivery, and found that the TRPV4-KCa2.3 interaction localization with caveolin-1 in ECs decreased (Appendix Fig S7). Furthermore, exposure to methyl β-cyclodextrin, an agent known to disrupt caveolae and caveolae-dependent signaling (Lohn et al, 2000; Dedkova et al, 2003), decreased the TRPV4-KCa2.3 interaction and GSK1016790A-induced $K^+$ efflux in primary ECs (Appendix Fig S8A and B), as well as GSK1016790A- or ACh-induced vasodilation

in freshly isolated arterial segments (Appendix Fig S8C). Consistently, knockdown of caveolin-1 in ECs by adeno-associated virus (AAV) siRNA (AAV-siRNA-caveolin-1) (Morishita et al, 1995) decreased the TRPV4-KCa2.3 interaction and GSK1016790A-induced $K^+$ efflux in primary ECs, as well as GSK1016790A- or ACh-induced vasodilation in freshly isolated arterial segments (Appendix Fig S8D). These results suggest that TRPV4 and KCa2.3 cluster in specific positions within the cell membrane to form functional units and that caveolae may constitute the scaffolding for this micro-compartment organization.

## Increasing the TRPV4-KCa2.3 interaction lowers blood pressure, but dissociation of TRPV4-KCa2.3 interaction increases blood pressure

To investigate the role of TRPV4-KCa2.3 interaction in the regulation of blood pressure in vivo, a KCa2.3-expressing AAV vector (AAV-Flt1-KCa2.3) was used to increase the interaction in the ECs of normotensive mice. Intravenously injected AAV-Flt1-KCa2.3 effectively infected the endothelium (Fig 2A) and significantly increased the degree of TRPV4-KCa2.3 co-localization in vivo (Fig 2B). Functionally, AAV-Flt1-KCa2.3 enhanced the GSK1016790A-induced $K^+$ efflux from primary ECs (Fig 2C), as well as GSK1016790A-induced or ACh-induced vasodilation in freshly isolated arterial segments (Fig 2D). As a result, AAV-Flt1-KCa2.3 significantly lowered the mean arterial pressure of normotensive mice (Fig 2E). Notably, AAV-Flt1-KCa2.3-enhanced $K^+$ efflux and vasodilation, along with the hypotensive effect of AAV-Flt1-KCa2.3, were absent from TRPV4-knockout mice (Fig 2F–H), suggesting that the functional interaction of TRPV4 and KCa2.3 depends on TRPV4 in order to modulate blood pressure.

We then generated an AAV vector that expressed the 17 amino acids in the C-terminal of KCa2.3 in ECs only, AAV-Flt1-KCa2.3-17C, to reduce TRPV4-KCa2.3 interaction in normotensive mice. In contrast to the findings with AAV-Flt1-KCa2.3, tail injection of AAV-Flt1-KCa2.3-17C into normotensive mice significantly decreased the degree of TRPV4-KCa2.3 interaction in vivo (Fig 2I). It also reduced the GSK1016790A-induced $K^+$ efflux from primary ECs (Fig 2J), as well as GSK1016790A-induced or ACh-induced vasodilation in freshly isolated arterial segments (Fig 2K). As a result, AAV-Flt1-KCa2.3-17C increased the mean arterial pressure (Fig 2L). Notably, these effects of AAV-Flt1-KCa2.3-17C were abolished in the absence of TRPV4 (Fig 2M–O).

**Figure 2.** **Increasing the TRPV4-KCa2.3 interaction lowers blood pressure, but dissociation of TRPV4-KCa2.3 interaction increases blood pressure.**

A–H Mice were intravenously injected with AAV-Flt1-KCa2.3 and the AAV-Flt1 vector served as a control. (A) Representative en face (left) and cross-sectional (right) fluorescence images of AAV-Flt1-KCa2.3 or vector (flag-tag) expression in AAV-Flt1-infected mice. Green, autofluorescence of elastin underneath the endothelium (excitation, 488 nm); red, anti-flag. Scale bars, 10 μm. (B) Seven mice were tested for the effects of AAV-Flt1-KCa2.3 or vector in immuno-FRET experiments ($^\#P < 0.0001$ vs. control, unpaired t-test). (C, F) $K^+$ efflux from primary cultured ECs ($^\#P < 0.0001$ vs. control, unpaired t-test). (D, G) Vasodilation assays by wire myography (GSK1016790A (GSK) and ACh in freshly isolated arterial segments) ($^*P = 0.047$, $^\#P < 0.0001$ vs. control, two-way ANOVA). (E, H) Time course of the changes in mean arterial pressure (ΔMAP) in wild-type (C–E) and TRPV4-knockout (KO) (F–H) mice after tail injection of AAV-Flt1-KCa2.3, AAV-Flt1 vector, or vehicle (injection solution).

I–O Mice were intravenously injected with AAV-Flt1-KCa2.3-17C; the AAV-Flt1 vector served as the control. (I) Mice were tested for effects of AAV-Flt1-KCa2.3-17C or vector with immuno-FRET experiments ($^\#P < 0.0001$ vs. control, unpaired t-test). (J, M) $K^+$ efflux from primary cultured ECs ($^\#P < 0.0001$ vs. control, unpaired t-test). (K, N) Vasodilation assays by wire myography (GSK1016790A (GSK) and ACh) in freshly isolated arterial segments) ($^*P = 0.004$, $^\#P < 0.0001$ vs. control, two-way ANOVA). (L, O) Time course of the changes in mean arterial pressure (ΔMAP) in wild-type (J–L) and TRPV4-knockout (KO) (M–O) mice after tail injection of AAV-Flt1-KCa2.3-17C, AAV-Flt1 vector, or vehicle (injection solution).

Data information: Data are mean ± SD.

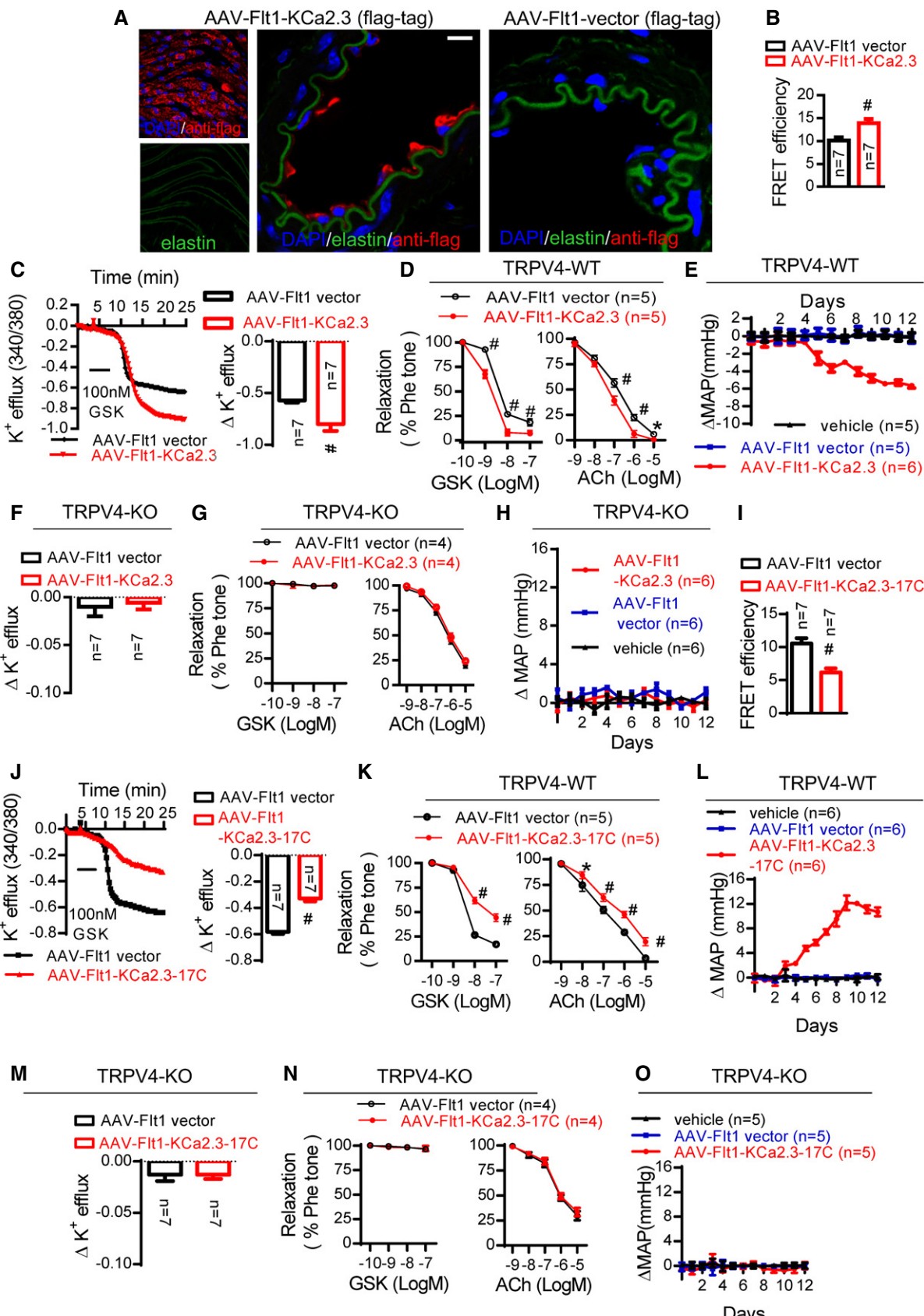

**Figure 2.**

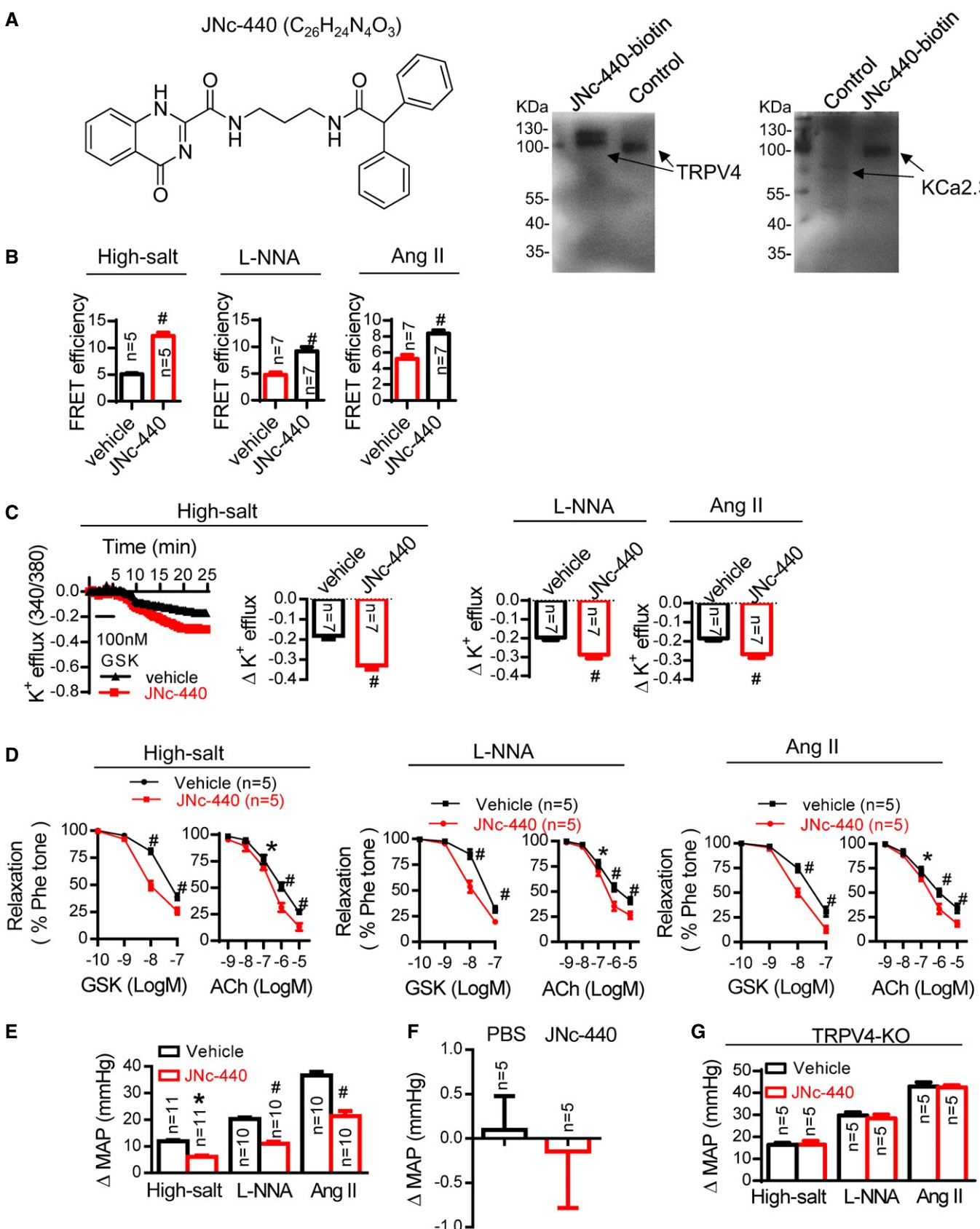

Figure 3.

**Figure 3.  Increasing the TRPV4-KCa2.3 interaction with JNc-440 is antihypertensive.**

A    Left: structural formula of JNc-440. Right: affinity of JNc-440 for both TRPV4 and KCa2.3. Biotinylated JNc-440 (JNc-440-biotin) pulled down TRPV4 or KCa2.3 in primary cultured ECs.

B–G  (B) Immuno-FRET experiments in arterial sections, (C) K$^+$ efflux from primary cultured ECs, (D) vasodilation assays by wire myography (GSK and ACh) in freshly isolated arterial segments, and (E) mean arterial pressure. (F) Effect of JNc-440 on blood pressure in normotensive mice. (G) Effect of JNc-440 (1 mg/kg; daily for 1 week) on blood pressure in TRPV4-knockout (KO) mice treated with a high-salt diet, L-NNA, or AngII. WT mice were treated with a high-salt diet, L-NNA, or AngII and then intravenously injected with JNc-440 (1 mg/kg) or vehicle for 2 h (B–F) or daily for 1 week (G). Data are mean ± SD. (B, C) $^#P < 0.0001$ vs. control, unpaired *t*-test. (D) *$P = 0.025$ (high-salt diet), *$P = 0.008$ (L-NNA treatment), or *$P = 0.021$ (AngII treatment), $^#P < 0.0001$ vs. control, two-way ANOVA. (E) *$P = 0.001$, $^#P < 0.0001$ vs. control, two-way ANOVA.

Source data are available online for this figure.

**Increasing the TRPV4-KCa2.3 interaction with JNc-440 is antihypertensive**

The results shown in Fig 2 indicate that manipulating the TRPV4-KCa2.3 interaction is effective in modulating arterial function and blood pressure. We therefore set out to develop a molecule to attach the AR2 region of TRPV4 onto the 17C region of KCa2.3 in ECs to restore the TRPV4-KCa2.3 dissociation found under hypertensive conditions (see strategy in Fig 1A). Based on an analysis of the molecular docking between the 3D structures of TRPV4 and KCa2.3, we screened and synthesized a series of chemical compounds. Among these, JNc-440 ($C_{26}H_{24}N_4O_3$, *N*-(3-(2,2-diphenylacetamido) propyl)-4-oxo-1,4-dihydroquinazoline-2-carboxamide) was a potent enhancer of TRPV4-KCa2.3 interaction because it showed high affinity for both TRPV4 and KCa2.3 (Fig 3A and Appendix Fig S9). In the setting of a high-salt diet, JNc-440 had a protective effect on TRPV4-KCa2.3 interaction *in vitro* at 10 μM (Appendix Fig S10A). Next, JNc-440 was injected into the tails of mice treated with a high-salt diet, L-NNA intake, or AngII delivery to further assess its protective effect on the physical and functional interaction of TRPV4-KCa2.3 and blood pressure. After injection, JNc-440 circulated in the blood (Appendix Fig S10B) and showed a clear distribution throughout the ECs 15 min after injection (Appendix Fig S10C), as well as significantly augmenting the degree of TRPV4-KCa2.3 interaction in the three types of hypertensive mice (Fig 3B). This occurred as early as 2 h after injection at 1 mg/kg (Appendix Fig S10D). Treatment with JNc-440 also enhanced the GSK1016790A-induced K$^+$ efflux in ECs (Fig 3C) and increased GSK1016790A- or ACh-induced vasodilation in freshly isolated arterial segments (Fig 3D) from hypertensive mice. Further, JNc-440 had significant antihypertensive effects (Appendix Fig S10E; Fig 3E, daily for 1 week). Importantly, the antihypertensive effect was absent in normotensive mice (Fig 3F and Appendix Fig S10F). Further, these effects depended on the presence of TRPV4, as JNc-440 was ineffective in TRPV4-knockout mice (Fig 3G). These findings suggest that JNc-440 increases the impaired TRPV4-KCa2.3 interaction to improve endothelium-dependent relaxation in small resistance arteries and to lower blood pressure.

Endothelial TRPV4 and KCa2.3 are effective targets for modulating blood pressure (Taylor *et al*, 2003; Brahler *et al*, 2009; Earley *et al*, 2009; Gao & Wang, 2010). As TRPV4 and KCa2.3 are both widely expressed and participate in important physiological functions, drugs that cause their systemic activation often have severe adverse effects, including endothelial failure and circulatory collapse (Willette *et al*, 2008). We therefore assessed TRPV4 and KCa2.3 activity in normotensive mice treated with JNc-440 and found that JNc-440 did not alter the activity of either TRPV4 or KCa2.3 (Appendix Fig S11A and B). As a result, JNc-440 did not change the blood pressure in normotensive mice. Thus, we conclude that by strengthening impaired endothelial TRPV4-KCa2.3 interaction, JNc-440 might be a useful antihypertensive treatment without adverse systemic effects.

In summary, we have shown that TRPV4-KCa2.3 interaction in ECs is essential to modulate blood pressure, and JNc-440 can protect this interaction and treat hypertension. Our data highlight the importance of impaired TRPV4-KCa2.3 interaction in the progress of hypertension and provide a novel strategy of developing new antihypertensive drugs with fewer adverse effects.

# Materials and Methods

## Human artery specimens

The use of human arteries was approved by the Review Board at the Affiliated Hospital, Jiangnan University, and the experiments conformed to the principles set out in the WMA Declaration of Helsinki and the Department of Health and Human Services Belmont Report. All patients provided written informed consent for the collection of samples. Mesenteric, femoral, popliteal, and brachial arteries were harvested from surgical specimens from normotensive and hypertensive patients. The mean age of patients was 51.3 years (range, 41–62 years). A history of hypertension was defined as having persistent elevated blood pressure, systolic pressure > 140 mmHg, or diastolic pressure > 90 mmHg and requiring medical treatment. The background information on patients is presented in Appendix Table S1.

## Animals

All animal experiments were performed in accordance with the laboratory animal guidelines and with the approval of the Animal Experimentations Ethics Committee, Jiangnan University. TRPV4-knockout (TRPV4$^{-/-}$) (Suzuki *et al*, 2003) mice were from Dr. Makoto Suzuki (Jichi Medical School, Tochigi, Japan and RIKEN BioResource Center, Japan). TRPV4$^{-/-}$ and wild-type C57BL/6J mice at 2–4 months of age were used. In some experiments, mice were made hypertensive by (i) addition of the NOS inhibitor N$^{ω}$-nitro-L-arginine (L-NNA) to the drinking water (0.5 g/l; 7-day duration) (Earley *et al*, 2009; Gonzalez-Villalobos *et al*, 2013); (ii) delivery of AngII at an infusion rate of 500 ng/kg/min for 14 days by osmotic pump (Model 2004, Azlet); or (iii) provision of a high-salt diet (8% NaCl) for 3 weeks (Wang *et al*, 2012; Chen *et al*, 2015). A normal-salt diet (0.3% NaCl) diet served as control. Systolic and diastolic blood pressure was recorded by tail-cuff plethysmography

(NIPB-2 blood pressure monitor, Columbus Instruments), and the mean arterial pressure was calculated from these data as previously reported (Taylor *et al*, 2003; Gonzalez-Villalobos *et al*, 2013). The average of three to four measurements on each day of treatment was taken to be the representative pressure for each animal.

## Cell preparation and culture

Primary mesenteric artery endothelial cells were isolated from C57BL/6 mice and used for experiments without passage. Briefly, after anesthesia, the abdomen was opened, the small intestines were dissected out, and the mesenteric bed containing blood vessels along the small intestines was carefully removed. The remaining arterial branches were digested with 0.02% collagenase in endothelium basal medium for 45 min at 37°C. After centrifugation at 845 *g* for 5 min, the pelleted cells were re-suspended in endothelial complete medium (ECM) supplemented with 5% fetal bovine serum (FBS) and 1% endothelial cell growth supplement and cultured in a flask at 37°C with 5% $CO_2$. Non-adherent cells were removed 2 h later. The adherent endothelial cells were cultured with ECM at 37°C with 5% $CO_2$ for 1–2 days (Ma *et al*, 2013). HEK293 cells were cultured in Dulbecco's modified Eagle's medium supplemented with 10% FBS, 100 μg/ml penicillin, and 100 U/ml streptomycin.

## Immunostaining and immunofluorescence resonance energy transfer

Immuno-FRET was measured as previously described (Adebiyi *et al*, 2010). Tissue slides were de-paraffinized with 4% paraformaldehyde, washed with PBS, and incubated for 1 h with 10% bovine serum albumin and 1% Triton X-100 in PBS. The slides were incubated with the primary antibody [goat monoclonal anti-TRPV4 (Santa Cruz, SC-47527), rabbit polyclonal anti-KCa2.3 (Alomone labs, APC-025), or mouse monoclonal anti-CD31 (Abcam, ab9498), each at a dilution of 1:100 in PBS containing 5% BSA] overnight at 4°C in a humidified chamber. After washing with PBS, slides were incubated with secondary antibodies for 1 h at 37°C. After washing and mounting, fluorescence images were acquired using a confocal microscope (Leica TCS SP8). Acceptor photobleaching FRET measurements were carried out on the TRPV4-CFP+KCa2.3-YFP samples and the positive and negative controls with the confocal microscope. FRET efficiency results with the positive control (CFP-YFP fusion protein with an RNPPVAT-linker) were used as the reference for ratiometric FRET measurements. CFP was excited by the 458-nm line of an argon ion laser and detected between 465 and 510 nm; YFP was excited at 514 nm and detected between 535 and 590 nm. First, images of the donor and acceptor distributions were captured. Then, the acceptor dyes were bleached by repetitive scans with the 514-nm laser at maximal power. After photobleaching, a donor image was recorded again. The LSM data acquisition software (Leica TCS SP5 LAS AF version 1.7.0) was used to define regions of interest and calculate the mean cellular fluorescence intensity in each channel. Patients with hypertension and normal individuals were enrolled in our study at the Wuxi People's Hospital. Vascular samples from these study participants were dehydrated with sucrose solution, embedded with optimum cutting temperature compound, and stored at −80°C.

## Stochastic optical reconstruction microscopy (STROM) measurements

Cells were plated on confocal dishes at ~30% confluence. After 24 h, the cells were fixed in 3% paraformaldehyde and 0.1% glutaraldehyde in PBS, followed by washing with 0.1% sodium borohydride in PBS. Cells were blocked and permeabilized in blocking buffer (3% BSA with 0.2% Triton X-100 in PBS), incubated overnight at 4°C with primary antibody, washed three times, and then incubated for 45 min at room temperature with the secondary antibody. The primary antibodies used were goat anti-TRPV4 (sc-47527, Santa Cruz) and rabbit anti-KCa2.3 (APC-025, Alomone). The stains were Alexa Fluor 647 and Alexa Fluor 488 (Invitrogen). Spectrally resolved single-molecule imaging was performed in standard STROM imaging buffer that contained 5% (w/v) glucose, 100–200 mM cysteamine, 0.8 mg/ml glucose oxidase, and 40 μg/ml catalase, in Tris–HCl (pH 7.5 or 8.0). Approximately 4 μl of imaging buffer was dropped at the center of a freshly cleaned confocal dish (Xu *et al*, 2012).

## Whole-cell patch clamp

Whole-cell current was measured with an EPC-9 patch clamp amplifier. The pipette solution contained (in mmol/l): CsCl, 20; cesium-aspartate, 100; $MgCl_2$, 1; ATP, 4; $CaCl_2$, 0.08; 1,2-bis(o-aminophenoxy)ethane-N,N,N′,N′-tetraacetate, 10; and Hepes, 10 (pH, 7.2). Bath solution contained (in mmol/l): NaCl, 150; CsCl, 6; $MgCl_2$, 1; $CaCl_2$, 1.5; glucose, 10; and Hepes, 10 (pH, 7.4). Cells were clamped at 0 mV. Whole-cell current density (pA/pF) was recorded in response to successive voltage pulses of +80 and −80 mV for 100-ms duration and then plotted against time.

## Screen and synthesis of JNc-440

JNc-440 was screened as follows: (i) Preparation of receptor molecules: The crystal structural data (PDBNO, 4DX1) of TRPV4 were downloaded from the PDB database. We used DS to open 4DX1 and removed water molecules in the system view. With the help of the Clean Protein tool, hydrogen was added to the protein molecules; the non-standard name, the incomplete amino acids, and residues with more than one conformation were corrected. At the same time, the three-dimensional structure of the KCa2.3 protein was constructed using the homology modeling method. (ii) Preparation of ligand molecules: In this experiment, the ligand file used for molecular docking mainly came from the following four databases: ibs2010-mar-sc-upd, ibs2010mar-sc1, ibs2010mar-sc2, and TimTec-STOCK-2010V02. (iii) Definition of the active site of the receptor protein: We selected 4DX1, expanded Receptor-Ligand Interaction/Define and Edit Binding Site, in the tool browser and defined 4DX1 as the receptor molecule by clicking Receptor Define. According to the LibDock docking program, KCa2.3 and 4DX1 were defined as receptors with which molecular docking was carried out with the databases ibs2010mar-sc-upd, ibs2010mar-sc1, ibs2010mar-sc2, and TimTec-STOCK-2010V02. Taking 4DX1 as an example, we selected Ligand in the system view and Edit in the menu bar; then we clicked Select and opened its dialog box, choosing the Radius while setting the rest as Default, and clicked Apply. The AR2 amino acid sequence of the conserved structure of TRPV4 is (TDEEFREPS)TGKTCLPKALLNL SNGRNDTIPVLLDIAERTGNMREFINSPFRDIYY (aa 190–236). The

17C region of KCa2.3 is RKLELTKAEKHVHNFMM (aa 545–561). Selecting these amino acids, we expanded the Receptor-Ligand Interaction/Define and Edit Binding Site in the tool browser and defined a red ball at the site of conjunction by clicking From Current Selection; meanwhile, we customized the SBD-Site-Sphere in the system view. Selecting the SBD-Site-Sphere and right-clicking Attribute of SBD-Site-Sphere, we changed the radius of the ball to 12. (iv) Using the Dock Ligands (LibDock) program for molecular docking: After clicking Protocols and then Dock Ligands (LibDock) under Receptor-Ligand Interactions, the parameter setting window of Dock Ligands (LibDock) appeared. Selecting User Specified from the drop-down list, the value of Max Hits to save was set to "10", and other parameters were set to default values. Then, we clicked "run" in the toolbar to perform molecular docking. (v) Molecular docking results: It was evident that each docking produced a large number of conformations, many of which were different conformations of the same molecule, in other words, inconvenient for observation and calculation. We kept only the conformation with the highest value in LibDockScore so that each molecule successful in docking with a protein could only have one conformation. Using LibDockScore to score and sort, we compared the conformations of two receptor proteins with scores > 100. The same molecular with the highest average score was determined (Rao et al, 2007).

JNc-440 was synthesized as follows: (i) Compound **1** propylene diamine (30 g, 405 mmol, 1 eq) was dissolved in 150 ml dichloromethane and stirred in an ice bath. Di-tert-butyl dicarbonate (16.1 g, 73 mmol, 0.18 eq) was dissolved and diluted in 50 ml dichloromethane. The mixture was added slowly to the flask and then stirred at room temperature for 3 h. When a complete reaction was detected using thin-layer chromatography (TLC), the reaction solution was diluted with 50 ml dichloromethane and washed repeatedly with water, then washed with saturated NaCl solution, and dried over anhydrous $Na_2SO_4$. Compound **2** (19 g, 27%) was obtained after concentration. (ii) Compound **2** (10 g, 57 mmol, 1 eq) and triethylamine (8.7 g and 86 mmol, 1.5 eq) were dissolved in 100 ml dichloromethane and stirred in an ice bath. Diphenyl acetyl chloride (13.1 g, 57 mmol, 1 eq) was dissolved in 30 ml dichloromethane by slow addition to the flask and stirred at room temperature for 2.5 h. When a complete reaction was detected using TLC, the crude product was obtained after concentration. Column chromatography separation (20:1 dichloromethane/methanol) was conducted to obtain compound **3** (13.2 g, 63%). (iii) Compound **3** (8 g, 22 mmol, 1 eq) was dissolved in mixed solvent (40 ml, dichloromethane:trifluoroacetic acid = 4:1). The mixture was heated in an oil bath at 35°C for 1 h. When a complete reaction was detected using TLC, ammonia was used to adjust the pH of the reaction solution to 8–9 until solids were precipitated. Compound **4** (5.3 g, 90%) was obtained after filtration and drying. (iv) Compound **4** (2 g, 7 mmol, 2 eq) and 4-quinazoline ketone-2-formic acid ethyl (0.8 g, 3.5 mmol, 1 eq) were dissolved in 6 ml ethanol and put in a microwave reactor at 100°C for 45 min. When a complete reaction was detected using TLC, a white solid was precipitated from the reaction solution after natural cooling. Compound **5** (JNc-440) (0.7 g, 45%) was obtained after filtering and drying the solid. The structure of JNc-440 was confirmed with [1]H NMR (400 MHz, CDCl3): δ: 8.34 (d, J = 7.6 Hz, 1H), 8.20 (s, 1H), 7.76–7.81 (m, 2H), 7.28–7.36 (m, 10H), 6.13 (t, 1H), 4.98 (s, 1H), 3.48 (q, J = 11.2 Hz, 2H), 3.38 (q, 10.2 Hz, 2H), 1.81 (m, J = 12.4 Hz, 2H). MS (ESI) m/z: 441.3 (M+H)[+].

To synthesize biotinylated JNc-440, the following procedures were performed. (v) Compound **5** (30 mg, 0.068 mmol, 1 eq) was dissolved in mixed solvent (0.5 ml, acetonitrile:toluene = 1:1) and stirred, and phosphorus oxychloride (2 drops) was dissolved in mixed solvent (0.25 ml, acetonitrile:toluene = 1:1) by slow addition to the flask and stirred at 40°C. DIPEA (3 drops) was dissolved in mixed solvent (0.25 ml, acetonitrile:toluene = 1:1) by slow addition to the flask and stirred at 70°C for 3 h. When a complete reaction was detected using TLC, compound **6** was obtained after rotary evaporation at 55°C (25 mg, 80.1%). (vi) Compound **6** (25 mg, 0.055 mmol, 1 eq), triethylamine (6.628 mg, 0.066 mmol, 1.2 eq), and 2-(4-aminophenyl) ethanol (8.986 mg, 0.065 mmol, 1.2 eq) were dissolved in 1 ml isopropyl alcohol. The mixture was heated in an oil bath at 45°C for 2 h. When a complete reaction was detected using TLC, the crude product was obtained after concentration. Column chromatography separation (20:1 dichloromethane/methanol) was conducted to obtain compound **7** (20 mg, 65.6%). (vii) Compound **7** (20 mg, 0.036 mmol, 1 eq) and vitamin H (10.5 mg, 0.043 mmol, 1.2 eq) were dissolved in 2 ml DMF and stirred at 45°C. DMAP (0.874 mg, 0.007 mmol, 0.2 eq) was added to the flask. CDI (6.962 mg, 0.043 mmol, 1.2 eq) was dissolved in DMF (2 ml) by slow addition to the flask and stirred at 50°C for 3 h. When a complete reaction was detected using TLC, the crude product was obtained after concentration. Column chromatography separation (2:1 petroleum ether:ethyl acetate) was conducted to obtain compound **8** (biotinylated JNc-440, 6 mg, 21.4%). The structure of biotinylated JNC-440 was analyzed by [1]H NMR (400 MHz, CDCl3): δ: 8.34 (d, J = 7.6 Hz, 1H), 8.20 (s, 1H), 7.76–7.81 (m, 2H), 7.38 (m, 2H), 7.02 (m, 2H), 7.28–7.36 (m, 10H), 6.13 (t, 1H), 4.98 (s, 1H), 4.42 (m, 2H), 3.45 (t, 2H), 3.38 (t, 2H), 3.35 (m, 1H), 2.87 (t, 2H), 2.83 (d, 2H),2.16–2.08 (m, 4H), 2.30 (m, 2H), 1.91 (m, 2H). 1.69–1.62 (m, 4H), 1.26 (m, 2H). MS (ESI) m/z: 784.3 (M+H)[+].

### Biotinylated JNc-440 pull-down

Primary mesenteric artery ECs were isolated from C57BL/6 mice, and then cultured cells were treated with biotinylated JNc-440 (10 μM) for 96 h under a 5% $CO_2$ atmosphere at 37°C. Cells were harvested and lysed with RIPA buffer. For precipitation of biotinylated proteins, 10 μl beads were incubated with cell lysates. Magnetic streptavidin beads (Thermo Fisher scientific, Waltham, MA, USA) were washed three times with lysis buffer. Beads were incubated overnight at 4°C on a rotating platform. After incubation, beads were washed three times with cell lysis buffer, and bound proteins were eluted by boiling for 5 min with 50 μl SDS sample buffer. Samples were separated on an SDS–PAGE gel and transferred to polyvinylidene fluoride membranes. The membranes were incubated overnight with primary antibodies (anti-TrpV4 and anti-KCa2.3) at 4°C. Immunodetection was accomplished using a horseradish peroxidase-conjugated secondary antibody (1:3,000) and an enhanced chemiluminescence detection system (GE Healthcare).

### Endothelial cell-specific adeno-associated viruses

To produce EC-specific AAVs, the TRPV4, TRPV4-AR2, and KCa2.3-17C genes were sub-cloned into pAOV.SYN.3FLAG plasmids to produce pAOV.SYN.TRPV4.3FLAG, pAOV.SYN.TRPV4-AR2.3FLAG, and pAOV.SYN.KCa2.3-17C.3FLAG (Neuron Biotech Co., Ltd,

Shanghai, China), respectively, which had an EC-specific promoter, flt1. AAV-TRPV4, AAV-TRPV4-AR2, and AAV-KCa2.3-17C were produced by transfection of AAV-293 cells with pAOV. SYN.TRPV4.3FLAG, pAOV.SYN.TRPV4-AR2.3FLAG, and pAOV. SYN.KCa2.3-17C.3FLAG, respectively, with AAV helper plasmid (pAAV Helper) and AAV Rep/Cap expression plasmid. Viral particles were purified by iodixanol step-gradient ultracentrifugation. The genomic titer was $2.5–3.5 \times 10^{12}$ genomic copies per ml as determined by quantitative PCR (Wu $et$ $al$, 2000).

## Measurement of intracellular $K^+$ and $Ca^{2+}$

Intracellular levels of $K^+$ were assessed by determining $K^+$-binding benzofuran isophthalate (PBFI) fluorescence (Invitrogen P-1267MP) (Kozoriz $et$ $al$, 2010). Cells were equilibrated with the cell-permeant acetoxylmethyl ester of the ion-sensitive dye PBFI (5 μM) for 1 h and then washed and re-suspended in normal physiological saline solution (NPSS). Fluorescence was measured using dual excitation at 340 and 380 nm on an Olympus fluorescence imaging system. Changes in $[K^+]_i$ were calculated as the change in the PBFI ratio. $[Ca^{2+}]_i$ in cultured cells was measured with 10 μM Fura-4/AM and 0.02% pluronic F-127 for 1 h in the dark at 37°C in NPSS using the Leica TSC SP8 confocal microscope.

## Plasmids and mutations

For single-site mutation, deletion, or insertion, a PCR of 50 μl contained 2–10 ng template, 25 μl Prime STAR, and 0.5 μl of each of the two primer pairs. The PCR cycles were initiated at 95°C for 2 min to denature the template DNA, followed by 19 amplification cycles. Each cycle consisted of 95°C for 30 s, 56°C for 30 s, and 68°C for 5 min. The PCR cycles ended with an extension step at 68°C for 10 min. The PCR products were treated with five units DpnI at 37°C for 3 h, and then 10 μl of each PCR was analyzed by agarose gel electrophoresis. An aliquot of 2 μl of the above PCR products, the PCR products generated using QuickChange™, or those generated were each transformed into DH5α-competent cells. The transformed cells were spread on a Luria-Bertani plate containing antibiotics and incubated at 37°C overnight. The number of colonies was counted and used as an indirect indication of PCR amplification efficiency. Four colonies from each plate were grown, and the plasmid DNA was isolated. The primers were as follows: for AR1, forward: 5′-aagctcctgcacccacggggaagacctg-3′, reverse: 5′-caggtcttccccgtgggtgcaggagctt-3′; for AR2, forward: 5′-cggggagccgtcccgaggccagaca-3′, reverse: 5′-tgtctggcctcgggacggctcccg-3′; for AR3, forward: 5′-cagagacatctactactttggggagctgccct-3′, reverse: 5′-agggcagctccccaaagtagtagatgtctctg-3′; for AR4: forward: 5′-gggaggctacttctacaggggggaacacggtg-3′, reverse: 5′-caccgtgttcccctgtagaagtagcctccc-3′; for AR5: forward: 5′-ggcgacaggactcggatggcctttcgcc-3′, reverse: 5′-ggcgaaaggccatccgagtcctgtcgcc-3′;for AR6: forward: 5′-acctggagacagttctcaacaatgatgaggacaccc-3′, reverse: 5′-gggtgtcctcatcattgttgagaactgtctccaggt-3′; for TRPV4-CaMBD: forward: 5′-ctaccagtactatggcttcgagctgaacaagaactcaa-3′, reverse: 5′-ttgagttcttgttcagctcgaagccatagtactggtag-3′; for 17C: forward: 5′-tggtgagctgagtgtcaaccacagctaccacaa-3′, reverse: 5′-ttgttggtagctgtggttgacactcagctcacca-3′; for α1: forward: 5′-catgtgcacaacttcatgatgctaaagaagattgaccatgcc-3′, reverse: 5′- ggcatggtcaatcttctttagcatcatgaagttgtgcacatg-3′; for loop: forward: 5′-gtctaaaacatacaaagctgcatgccaaagtcaggaaacac-3′, reverse: 5′-gtgtttcctgact

ttggcatgcagctttgtatgtttatagac-3′; for α2: forward: 5′-aaagctgctaaagaagattgacatgtatgacttaatcacggagc, reverse: 5′-gctccgtgattaagtcatacatgtcaatcttctttagcagcttt-3′; for KCa2.3-CaMBD: forward: 5′-ctccgtgattaagtcatacatcatcatgaagttgtgcacatg-3′, reverse: 5′-catgtgcacaacttcatgatgatgtatgacttaatcacggag-3′; and for KCa2.3-17C-CaMBD: forward: 5′-ccgtgattaagtcatacataaccacagctaccacaaggg-3′, reverse: 5′-cccttgtggtagctgtggttatgtatgacttaatcacgg-3′. Sequences were verified by PCR (Garcia-Elias $et$ $al$, 2008).

## Arterial tension measurement

To measure changes in the isometric tension of arteries, 2-mm rings of mouse mesenteric artery were mounted in a myograph (Model 610M, Danish Myo Technology, Aarhus, Denmark) and bathed in oxygenated (95% $O_2$–5% $CO_2$) Krebs solution maintained at 3°C (pH ~7.4). The rings were stretched to an optimal baseline force of 1 mN. After 60 min of equilibration at baseline tension, 60 mM KCl solution (the NaCl in Krebs solution was substituted with an equimolar amount of KCl) was used to confirm the viability of the artery. Arterial rings were contracted with 1 μM phenylephrine. Once a steady-state tension was obtained, agonists were added to the bath in a cumulative fashion to obtain concentration–response curves. The bath solution was modified Krebs solution (Zhang $et$ $al$, 2014a).

## Plasma and EC sample preparation and HPLC-MS/MS analysis

To a 200 μl aliquot of rat plasma, 600 μl acetonitrile was added. After vortexing the mixture for 30 s and centrifugation at 13,523 $g$ for 8 min, the clear supernatant was transferred into a vial, and 20 μl was injected for HPLC analysis. Each weighed tissue sample was thawed and then homogenized in ice-cold physiological saline (250 mg/ml). Then, 100 μl of tissue homogenate was pipetted and further treated like the plasma samples. The analysis of JNc-440 was carried out using a Waters 2695 HPLC system coupled with a Waters Quattro Micro™ triple quadrupole mass spectrometer (Waters, USA). The separation was carried out using a Waters XBridge C18 column (50 × 2.1 mm, 3.5 μm). The isocratic mobile phase consisted of acetonitrile–0.1% formic acid (10:90, v/v) with a flow rate of 0.3 ml/min. The temperature of the column was maintained at 35°C. The mass spectrometer was operated in the positive ion detection mode with capillary voltage 3,500 V, desolvation temperature 350°C, and source temperature 100°C. Ion monitoring was performed using the multiple reaction-monitoring mode in order to measure the following ion transitions: m/z441→m/z167 was used for quantitation of JNc-440 and m/z497→ m/z192 for the internal standard (Seger $et$ $al$, 2009).

## Statistics

ECs and mice were randomized into different treatment groups for each experiment, and all mice were used throughout the experiments unless they died. The response of animals to treatment was quantified by two investigators in a blinded fashion. No statistical methods were used to predetermine sample size. Sample sizes and animal numbers were chosen on the basis of previous publications and experiment types and are indicated in each figure. The results show the mean ± SD. Comparisons among groups were made using ANOVA or unpaired Student's $t$-test, with $P < 0.05$ as the threshold for a significant difference.

---

## The paper explained

### Problem

The currently available antihypertensive agents have undesirable adverse effects due to systemically changing target activity (receptors, channels, and enzymes). Changes in the vascular resistance of small resistance arteries have been implicated in the development of hypertension. However, the critical role of these arteries in the pathogenesis of hypertension has been ignored to date, making them potential targets to develop new methods to treat hypertension with less adverse effects.

### Results

With the goal of effectively treating hypertension and cutting down on the accompanying adverse effects due to systemic activation of target, we explored the mechanism of hypertension caused by dysregulated endothelial TRPV4-KCa2.3 interaction in small resistance arteries; we also identified a possible therapeutic small molecule that repairs impaired endothelial TRPV4-KCa2.3 interaction but did not systematically change TRPV4 and KCa2.3 activity.

### Impact

Our data highlight the importance of impaired TRPV4-KCa2.3 interaction in the progression of hypertension and provide a novel strategy for developing new antihypertensive drugs with fewer adverse effects.

**Expanded View** for this article is available online.

## Acknowledgements

We thank Prof Iain C Bruce for critical reading of the manuscript. This work was supported by the China National Natural Science Foundation (91439131, 81622007, an 81572940 to XM, 31550006 to DXH, and 21305051 to CLT); the National High-Technology Research and Development Program (863 Program) of China (2015AA020948 to XM); the Natural Science Foundation for Distinguished Young Scholars of Jiangsu Province (BK20140004 to XM); and Fundamental Research Funds for the Central Universities (JUSRP51704A and JUSRP51615B).

## Author contributions

QP, CS, DH, ZC, and PZ performed FRET, STORM, mutation, intracellular $K^+$ and $Ca^{2+}$, arterial tension, patch clamp and membrane potential experiments; DS and CT contributed to the design and synthesis of small-molecule compounds; AM, YZhu, CL, HL, and MX contributed to the tail injections, animal housing, and blood pressure experiments; YZhou performed HPLC-MS/MS analysis; ZY provided technical assistance and obtained clinical samples; JJ and XY provided a critical reading of the manuscript; DH and XM initiated the project, designed the study, and wrote the paper; and BN designed the study, discussed data and provided a critical reading of the manuscript.

## Conflict of interest

X.M., D.H., C.T., P.Z., Z.C., and Y.F.C. have filed a patent "A compound for use in increasing impaired TrpV4-KCa2.3 interaction and its application in anti-hypertension" to the State Intellectual Property Office of P.R. China (201710224218.9, PCT/CN2017/084079).

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
