## [Review Process File · EMBO Molecular Medicine]

Treatment of hypertension by increasing impaired endothelial TRPV4-KCa2.3 interaction

Dongxu He, Qiongxi Pan, Zhen Chen, Chunyuan Sun, Peng Zhang, Aiqin Mao, Yaodan Zhu, Hongjuan Li, Chunxiao Lu, Mingxu Xie, Yin Zhou, Daoming Shen, Chunlei Tang, Zhenyu Yang, Jian Jin, Xiaoqiang Yao, Bernd Nilius, Xin Ma

Corresponding author: Prof. Xin Ma, Jiangnan University

Review timeline:

Submission date:	20 February 2017
Editorial Decision:	12 April 2017
Revision received:	31 July 2017
Editorial Decision:	14 August 2017
Revision received:	18 August 2017
Accepted:	18 August 2017

Transaction Report:

Editor: Roberto Buccione

1st Editorial Decision

12 April 2017

Thank you for the submission of your manuscript to EMBO Molecular Medicine. We have now heard back from the Reviewers whom we asked to evaluate your manuscript.

I again apologise for the unusual delay in reaching a decision on your manuscript. In this case, we first experienced significant difficulties in securing expert and willing reviewers. I eventually only managed to secure two reviewers. Further to this, the evaluations were delivered with some delay.

I am therefore proceeding based on the two evaluations obtained so far as further delay cannot be justified and would not be productive.

As you will see, although the reviewers find your work potentially interesting and relevant, they both point to important technical and formal. For instance reviewer 2 notes two fundamental points, i.e. lack of direct experimental support that TRPV4 and KCa2.3 interact and lack of understanding of how JNC-440 performs compared with other anti-hypertensives.

Our reviewer cross-commenting exercise confirmed the positive stance but also the need to address the issues as a requirement for publication.

In conclusion, while publication of the paper cannot be considered at this stage, given the potential interest of your findings and after internal discussion, we have decided to give you the opportunity to address the criticisms.

We are thus prepared to consider a substantially revised submission, with the understanding that the Reviewers' concerns must be addressed with additional experimental data where appropriate and as outlined above, and that acceptance of the manuscript will entail a second round of review. The overall aim is to significantly upgrade the relevance and conclusiveness of the dataset, which of course is of paramount importance for our title.

Please note that it is EMBO Molecular Medicine policy to allow a single round of revision only and that, therefore, acceptance or rejection of the manuscript will depend on the completeness of your responses included in the next, final version of the manuscript.

EMBO Molecular Medicine now requires a complete author checklist (<http://embomolmed.embopress.org/authorguide#editorial3>) to be submitted with all revised manuscripts. Provision of the author checklist is mandatory at revision stage; the checklist is designed to enhance and standardize reporting of key information in research papers and to support reanalysis and repetition of experiments by the community. The list covers key information for figure panels and captions and focuses on statistics, the reporting of reagents, animal models and human subject-derived data, as well as guidance to optimise data accessibility. This checklist especially relevant in this case given the issues raised with respect to statistical treatment and animal numbers.

As you know, EMBO Molecular Medicine has a "scooping protection" policy, whereby similar findings that are published by others during review or revision are not a criterion for rejection. However, I do ask you to get in touch with us after three months if you have not completed your revision, to update us on the status. Please also contact us as soon as possible if similar work is published elsewhere.

We now mandate that all corresponding authors list an ORCID digital identifier. You may acquire one through our web platform upon submission and the procedure takes <90 seconds to complete. We also encourage co-authors to supply an ORCID identifier, which will be linked to their name for unambiguous name identification.

Last, but not least, please carefully conform to our author guidelines (<http://embomolmed.embopress.org/authorguide>) to ensure rapid pre-acceptance processing in case of a favourable outcome on your revision.

I look forward to seeing a revised form of your manuscript in due time.

***** Reviewer's comments *****

Referee #1 (Remarks):

The authors addressed to the function of small arteries in terms of endothelia K⁺ channels and EDHF signaling to treat hypertension. They identified a reduced interaction between TRPV4 and KCa2.3 in EC from hypertensive patients and murine hypertensive model. They further developed a small-molecule drug, JNc-440, which strengthened the TRPV4 and KCa2.3 coupling in ECs and enhanced vasodilation and exerted antihypertensive effects in mice. These results are interesting to explore the novel antihypertensive drug. Our concerns are as follows.

1. In Figure 1g, ACh-induced vasodilation and sodium nitroprusside-induced vasodilation should be shown to clarify the EDHF or NO-dependent vasodilation.
2. In terms of the results from ACh-induced vasodilation in several experiments, the results are similar to that from GSX-induced vasodilation in hypertensive mice model, but not in TRPV4-KO mice (Figure 3C and 3f). How could you explain about the discrepancy?
3. Similarly, in Figure 4d, because L-NNA is known as an inhibitor for NO, the reviewer would like to know whether the effect of Ach is NO-independent or not. In Fig. S5c, the effect of caveolin-1 colocalization also affect Ach-dependent vasodilation.
4. In Figure 4e-g, the mean arterial blood pressure was shown in each group. Although the time course of this data is one week after venous injection, could you show the acute effect of JNc-440 in blood pressure? They demonstrated that the amount of TRPV4-KCa2.3 coupling is observed as early as 2 hours after venous injection. Thus, the time course of two experiment is quite different.

How long is the effect of JNc-440 sustained?

5. Is there any difference of the basal level of blood pressure between wild type and TRPV4-KO mice? In the results of blood pressure, the response to high salt, L-NNA and AngII in TRPV4-KO is same with that in wild type. It suggests that TRPV4-KCa2.3 does not contribute the increase in blood pressure. How could you explain that?

Referee #2 (Remarks):

This work performed by Dongxu He and his colleagues uncovered a previous unrecognized connection between TRPV4 and KCa2.3, which was critical to the development of hypertension. Moreover, authors also provided an effective intervention drug Jnc-440, which augmented the TRPV4/KCa2.3 coupling to counteract high blood pressure without affecting their activities. This work was well-performed and written with novelty. However, some results are necessary be optimized to strengthen the conclusion. Here list some suggestions and comments.

1. Most of the TRPV4-KCa2.3 coupling results were presented using immunofluorescent staining (FRET). Some other experiments detecting the interaction between the two molecules, such as co-IP, should be also used to further support the critical conclusion that this coupling was a critical step in the process of hypertension. Also, the effect of caveolae should be also confirmed.

2. The positive FRET signal depends on the physical colocalization of TRPV4 and KCa2.3. However, it is not an appropriate method to prove the existence of functional coupling between the two channels. Thus, additional experiments showing the functional interaction of TRPV4 and KCa2.3 should be included to validate this point.

3. The reference describing cyclodextrin needs be added. In addition, the effect of cyclodextrin should be confirmed by adding AAV-siRNA-caveolae in primary ECs.

4. This work indicates that JNc-440 significantly strengthens the destroyed endothelial TRPV4-KCa2.3 coupling in hypertensive models. However, the authors did not show the effect of JNc-440 on normal physiological condition to prove the safety and specificity of this new drug. To emphasize its superiority, it is suggested that the anti-hypertensive effect of JNc-440 should be compared with other widely used anti-hypertensive drugs such as calcium channel antagonists and angiotensin II receptor blockers in discussion

1st Revision - authors' response

31 July 2017

Reviewer #1.

General Comments:

The authors addressed to the function of small arteries in terms of endothelia K⁺ channels and EDHF signaling to treat hypertension. They identified a reduced interaction between TRPV4 and KCa2.3 in EC from hypertensive patients and murine hypertensive model. They further developed a small-molecule drug, JNc-440, which strengthened the TRPV4 and KCa2.3 coupling in ECs and enhanced vasodilation and exerted antihypertensive effects in mice. These results are interesting to explore the novel antihypertensive drug. Our concerns are as follows.

Answer: Thank you for your generous comments.

Specific Comment #1) *In Figure 1g, ACh-induced vasodilation and sodium nitroprusside-induced vasodilation should be shown to clarify the EDHF or NO-dependent vasodilation.*

Answer: Thank you for your comments. We have now followed your suggestion and performed the experiments.

(1) In freshly isolated small arterial segments taken from mice that had been fed a high-salt diet, acetylcholine (ACh) induced relaxation was weakened, but sodium nitroprusside (SNP) relaxation was not significantly different (Figures for Referees not shown.). The results are consistent with previous reports¹ and suggesting the high-salt diet weakens EDHF-dependent, but not nitric oxide (NO)-dependent vasodilation.

(2) Consistently, membrane potentials were measured by sharp microelectrodes impaled from adventitial side in small arteries, we shown that the smooth muscle hyperpolarization, which is an indicator of EDHF release²⁻⁴, was activated by TRPV4 channel agonist GSK1016790A or ACh.

Antagonist of KCa2.3 (apamin) or TRPV4 (HC067047) inhibited ~80% of GSK1016790A or ACh-induced hyperpolarization (Figures for Referees not shown.). Thus, these data suggest EDHF is the dominant component here and TRPV4-KCa2.3 contributes to such EDHF signaling.

Specific Comment #2). *In terms of the results from ACh-induced vasodilation in several experiments, the results are similar to that from GSK-induced vasodilation in hypertensive mice model, but not in TRPV4-KO mice (Figure 3C and 3f). How could you explain about the discrepancy?*

Answer: TRPV4 channels play a major role in endothelial-dependent vasodilation⁵. Either GSK1016790A or ACh could activate TRPV4 channel. The former is synthetic, but specific^{6,7}; the latter is natural, but not specific. It is reported that, in TRPV4 KO mice, GSK1016790A-induced response was completely lost⁸, but ACh-induced response was impaired about 20-50%^{9,10}. Our data is consistent with previous reports. This is due to the different specificity of GSK1016790A and ACh to TRPV4 channel.

The before-mentioned information has been added into manuscript as discussion of fig. 2d and g (similar to fig. 3c compared to fig. 3f).

Specific Comment #3) *Similarly, in Figure 4d, because L-NNA is known as an inhibitor for NO, the reviewer would like to know whether the effect of ACh is NO-independent or not. In Fig. S5c, the effect of caveolin-1 colocalization also affect ACh-dependent vasodilation.*

Answer: Thanks for your comments. We followed your suggestions and performed additional experiments.

(1) In isolated small artery segments from L-NNA induced hypertensive mice, pre-incubation with nitric oxide synthase inhibitor L-NAME did not abolish ACh-induced (Figures for Referees not shown.). This result indicated that the effect of ACh is NO-independent. Previous reports showed that L-NNA delivery is a common animal model for hypertension^{11,12}. In this animal model, NO pathway was inhibited in vivo. Thus in small artery, compared to the role of NO, EDHF is dominant^{2,13}.

(2) The results in the previous version of manuscript, it is suggested that caveolin-1 colocalization also affect ACh-dependent vasodilation. In addition to pharmacological agent, we now further used AAV-siRNA-caveolin-1. We found that AAV-siRNA-caveolin-1, decreased TRPV4-KCa2.3 coupling and GSK1016790A-induced K⁺ efflux in primary ECs, as well as GSK1016790A- or acetylcholine (ACh)-induced vasodilation in freshly isolated arterial segments (Figures for Referees not shown.). The results are consistent with our former data.

Specific Comment #4) *In Figure 4e-g, the mean arterial blood pressure was shown in each group. Although the time course of this data is one week after venous injection, could you show the acute effect of JNc-440 in blood pressure? They demonstrated that the amount of TRPV4-KCa2.3 coupling is observed as early as 2 hours after venous injection. Thus, the time course of two experiments is quite different. How long is the effect of JNc-440 sustained?*

Answer: Thanks for your comments. We now followed your suggestions and performed additional experiments.

(1) We recorded blood pressure in time-course in vivo (0 h, 0.5 h, 1 h, 2 h, 3h, 4 h, 6h, 8 h, 12 h, 24 h) and found that JNc-440 showed markedly antihypertensive effects in high-salt, L-NNA and AngII-treated mice after tail injection at 1-2h (Figures for Referees not shown.), which sustained as long as 8-12 h, with a dose of 1 mg/kg.

We also recorded blood pressure in time-course in normotensive wild-type mice for 12 days (144 h) and no marked effect on blood pressure was observed (Fig. S10g). This result and results from fig. 4f together suggest antihypertensive effect of JNc-440 was absent in normotensive mice as long as 12 days of observation.

This information has been added into the manuscript.

(2) In the previous version of manuscript, we demonstrated that in high-salt, L-NNA and AngII-treated mice, JNc-440 showed significant antihypertensive effects one week after tail injection (Fig. 4e) with a minimum dose of 1 mg/kg (Fig. S7b). We made confusion here about different time course. It should be JNc-440 (q.d.; one week) still showed significant antihypertensive effects after tail injection. We made this modification in the current manuscript.

Specific Comment #5) *Is there any difference of the basal level of blood pressure between wild type and TRPV4-KO mice? In the results of blood pressure, the response to high salt, L-NNA and AngII*

in TRPV4-KO is same with that in wild type. It suggests that TRPV4-KCa2.3 does not contribute the increase in blood pressure. How could you explain that?

Answer: Thanks for your comments.

(1) We followed your suggestions and showed continuous resting MAPs (Figures for Referees not shown.). They were similar in TRPV4 KO and WT mice which consistent with previous reports^{8, 10, 14}.

(2) Although WT and TRPV4 KO animals became hypertensive after each treatment, our results and previous studies showed that the resulting level of blood pressure was greater in TRPV4 KO mice compared with WT controls, but not significant¹⁴.

We thought that in TRPV4 KO mice, there is a systematic change in these mice such as metabolic pathway and EC function, so that blood pressure regulation is complicated. During high salt, L-NNA and AngII treatment in TRPV4-KO mice, absence of TRPV4 may be compensated by other factors in circulation system to regulate blood pressure. In the further, we are planning to establish TRPV4/KCa2.3 knockout especial in EC, which could further explore the role of TRPV4/KCa2.3 in vivo.

Further, enlightened by the reviewer's question, we have tried to identify these factors by using metabonomics technology. We have now finished the primary metabonomics analysis in high salt, L-NNA and AngII treatment in WT and TRPV4-KO mice, and identified several factors may be involved. We are glad to share our new results in future.

Reviewer #2.

General Comments:

This work performed by Dongxu He and his colleagues uncovered a previous unrecognized connection between TRPV4 and KCa2.3, which was critical to the development of hypertension. Moreover, authors also provided an effective intervention drug Jnc-440, which augmented the TRPV4/KCa2.3 coupling to counteract high blood pressure without affecting their activities. This work was well-performed and written with novelty. However, some results are necessary be optimized to strengthen the conclusion. Here list some suggestions and comments.

Answer: Thank you for your generous comments.

Specific Comment #1) *Most of the TRPV4-KCa2.3 coupling results were presented using immunofluorescent staining (FRET). Some other experiments detecting the interaction between the two molecules, such as co-IP, should be also used to further support the critical conclusion that this coupling was a critical step in the process of hypertension. Also, the effect of caveolae should be also confirmed.*

Answer: Thanks for your comments. We followed your suggestions and performed additional experiments. We used co-IP to confirm the interaction of caveolin-1/TRPV4/KCa2.3.

(1) In results of co-IP experiments (Figures for Referees not shown.), an anti-KCa2.3 antibody could pull down TRPV4 proteins in the lysates freshly prepared from mice ECs. An anti-TRPV4 antibody could reciprocally pull down KCa2.3. An anti-caveolin-1 antibody could pull down either TRPV4 or KCa2.3 protein.

(2) Furthermore, in hypertensive models with a high-salt diet, L-NNA intake or AngII delivery, co-IP experiments found a reduced amount of TRPV4-KCa2.3 complex in small arterial ECs, even after KCa2.3 protein level was titrated to the same quantity in different loading lane (Figures for Referees not shown.).

These results have been added into manuscript.

Specific Comment #2) *The positive FRET signal depends on the physical colocalization of TRPV4 and KCa2.3. However, it is not an appropriate method to prove the existence of functional coupling between the two channels. Thus, additional experiments showing the functional interaction of TRPV4 and KCa2.3 should be included to validate this point.*

Answer: In the previous version of manuscript, we measured intracellular K^+ and arterial tension. TRPV4 allows Ca^{2+} influx, and KCa2.3 is Ca^{2+} sensitive K^+ channel which allow K^+ efflux. Thus, by activation of TRPV4 with specific TRPV4 agonist GSK1016790A, measurement of K^+ efflux and arterial tension could test the functional coupling.

Following to the reviewer's suggestion, we further performed whole-cell patch clamp to test the functional coupling of TRPV4-SKCa3. In primarily cultured ECs, TRPV4 activator GSK1016790A

induced a whole-cell current, which could be inhibited by TRPV4 inhibitor HC067047, but not by KCa2.3 inhibitor, apamin (Figures for Referees not shown.). In contrast, KCa2.3 activator, CyPPA, induced a whole-cell current, which could be inhibited by TRPV4 inhibitor, HC067047 (Figures for Referees not shown.). These results strongly supported that the functional coupling of TRPV4-KCa2.3 which also revealed by fluorescent K⁺ efflux measurement in the previous version of manuscript.

Specific Comment #3) *The reference describing cyclodextrin needs be added. In addition, the effect of cyclodextrin should be confirmed by adding AAV-siRNA-caveolae in primary ECs.*

Answer: (1) the reference describing methyl β -cyclodextrin (M β CD) was added in the current version of manuscript^{15, 16}.

(2) In addition to pharmacological agent, we further used AAV-siRNA-caveolin-1. We found that AAV-siRNA-caveolin-1 decreased TRPV4-KCa2.3 coupling and GSK1016790A-induced K⁺ efflux in primary ECs, as well as GSK1016790A- or acetylcholine (ACh)-induced vasodilation in freshly isolated arterial segments (Figures for Referees not shown.). The results are consistent with our former data.

Specific Comment #4) *This work indicates that JNc-440 significantly strengthens the destroyed endothelial TRPV4-KCa2.3 coupling in hypertensive models. However, the authors did not show the effect of JNc-440 on normal physiological condition to prove the safety and specificity of this new drug. To emphasize its superiority, it is suggested that the anti-hypertensive effect of JNc-440 should be compared with other widely used anti-hypertensive drugs such as calcium channel antagonists and angiotensin II receptor blockers in discussion.*

Answer: Thanks for your comments. We followed your suggestions and performed additional experiments. (1) We have shown that JNc-440 has no effect on blood pressure injected at 1 mg/kg/day in normotensive mice after one week of treatment (Fig. 4f). Here, we further showed the short-term effect of JNc400 on blood in normotensive mice within 24 h of treatment. Data showed that JNc-440 did not markedly affect blood pressure with intravenously injection at 1 mg/kg (Figures for Referees not shown.). We also recorded blood pressure in time-course in normotensive wild-type mice for 12 days (144 h) and no marked effect on blood pressure was observed (Fig. S11d). These results and results from fig. 4f together suggest antihypertensive effect of JNc-440 was absent in normotensive mice as long as 12 days of observation. This information has been added into the manuscript.

(2) The specificity of JNc-440 on normal physiological condition has shown as in normal physiological condition, JNc-440 (10 μ M) showed high affinity to both TRPV4 and KCa2.3 (Fig. 4a and Fig. S6) in mice ECs.

(3) We compared JNc-440 with other anti-hypertensive drugs in this manuscript mainly focus on the side effects: In addition to reducing blood pressure, currently available antihypertensive agents have undesirable adverse effects, due to the systemic actions of the drugs by systemically altering targets activity (receptors/channel/enzyme). These effects, such as loss of potassium ions for diuretics, bronchospasm for beta-blockers, constipation for Ca²⁺ channel blockers, and dry cough for ACEI, edema caused by unmatched circulation between arteries and veins during treatment of Ca²⁺ channel blockers¹⁷, lead to no adherence with therapies. We explored new antihypertensive method specifically targeting sites of dysfunction. We developed a small-molecule drug, JNc-440, which showed affinity to both TRPV4 and KCa2.3. JNc-440 significantly strengthened the TRPV4-KCa2.3 coupling in ECs and enhanced vasodilation and exerted antihypertensive effects in mice. Importantly, JNc-440 specifically targeted the destroyed TRPV4-KCa2.3 coupling in ECs but did not systemically activate TRPV4 and KCa2.3.

1. Barron, L.A., Giardina, J.B., Granger, J.P. & Khalil, R.A. High-salt diet enhances vascular reactivity in pregnant rats with normal and reduced uterine perfusion pressure. *Hypertension* 38, 730-735 (2001).
2. Busse, R. et al. EDHF: bringing the concepts together. *Trends in pharmacological sciences* 23, 374-380 (2002).
3. Feletou, M. & Vanhoutte, P.M. Endothelium-derived hyperpolarizing factor: where are we now? *Arteriosclerosis, thrombosis, and vascular biology* 26, 1215-1225 (2006).
4. Feletou, M. & Vanhoutte, P.M. EDHF: an update. *Clinical science* 117, 139-155 (2009).

5. Sonkusare, S.K. et al. Elementary Ca²⁺ signals through endothelial TRPV4 channels regulate vascular function. *Science* 336, 597-601 (2012).
6. Thorneloe, K.S. et al. N-((1S)-1-{{4-((2S)-2-{{(2,4-dichlorophenyl)sulfonyl}amino}-3-hydroxypropanoyl)-1-piperazinyl}carbonyl}-3-methylbutyl)-1-benzothiophene-2-carboxamide (GSK1016790A), a novel and potent transient receptor potential vanilloid 4 channel agonist induces urinary bladder contraction and hyperactivity: Part I. *The Journal of pharmacology and experimental therapeutics* 326, 432-442 (2008).
7. Thorneloe, K.S. et al. An orally active TRPV4 channel blocker prevents and resolves pulmonary edema induced by heart failure. *Science translational medicine* 4, 159ra148 (2012).
8. Willette, R.N. et al. Systemic activation of the transient receptor potential vanilloid subtype 4 channel causes endothelial failure and circulatory collapse: Part 2. *The Journal of pharmacology and experimental therapeutics* 326, 443-452 (2008).
9. Ma, X. et al. Functional role of TRPV4-KCa_{2.3} signaling in vascular endothelial cells in normal and streptozotocin-induced diabetic rats. *Hypertension* 62, 134-139 (2013).
10. Zhang, D.X. et al. Transient receptor potential vanilloid type 4-deficient mice exhibit impaired endothelium-dependent relaxation induced by acetylcholine in vitro and in vivo. *Hypertension* 53, 532-538 (2009).
11. Sollinger, D. et al. Damage-associated molecular pattern activated Toll-like receptor 4 signalling modulates blood pressure in L-NAME-induced hypertension. *Cardiovascular research* 101, 464-472 (2014).
12. Itani, H.A. et al. CD70 Exacerbates Blood Pressure Elevation and Renal Damage in Response to Repeated Hypertensive Stimuli. *Circulation research* 118, 1233-1243 (2016).
13. Edwards, G., Dora, K.A., Gardener, M.J., Garland, C.J. & Weston, A.H. K⁺ is an endothelium-derived hyperpolarizing factor in rat arteries. *Nature* 396, 269-272 (1998).
14. Earley, S. et al. TRPV4-dependent dilation of peripheral resistance arteries influences arterial pressure. *American journal of physiology. Heart and circulatory physiology* 297, H1096-1102 (2009).
15. Dedkova, E.N., Ji, X., Wang, Y.G., Blatter, L.A. & Lipsius, S.L. Signaling mechanisms that mediate nitric oxide production induced by acetylcholine exposure and withdrawal in cat atrial myocytes. *Circulation research* 93, 1233-1240 (2003).
16. Lohn, M. et al. Ignition of calcium sparks in arterial and cardiac muscle through caveolae. *Circulation research* 87, 1034-1039 (2000).
17. Moser, M. Antihypertensive medications: relative effectiveness and adverse reactions. *J Hypertens Suppl* 8, S9-16 (1990).

2nd Editorial Decision

14 August 2017

Thank you for the submission of your revised manuscript to EMBO Molecular Medicine. We have now received the enclosed reports from the reviewers that were asked to re-assess it. As you will see the reviewers are now supportive, although reviewer 2 has a few final requests, with which I agree (see also my point 7 below) that require your action.

I am thus prepared to accept your manuscript for publication pending compliance with reviewer 2's final requests and the following editorial requirements:

1) We encourage the publication of source data, with the aim of making primary data more accessible and transparent to the reader. Would you be willing to provide a PDF file per figure that contains the original, uncropped and unprocessed scans of all or at least the key gels used in the manuscript and/or source data sets for relevant graphs? The files should be labeled with the appropriate figure/panel number, and in the case of gels, should have molecular weight markers; further annotation may be useful but is not essential. The files will be published online with the article as supplementary "Source Data" files. If you have any questions regarding this just contact me.

2) Please note that I am suggesting some textual changes in the abstract and the title, which I would like you to consider. Please use the attached manuscript file for your amendments. Connected to this, the manuscript would benefit from language and text editing. You might consider one of the available language editing services e.g. <http://wileyeditingservices.com>.

For all the above, please refer to our author guidelines (<http://embomolmed.embopress.org/authorguide>).

Please submit your revised manuscript within two weeks. I look forward to seeing a revised form of your manuscript as soon as possible.

***** Reviewer's comments *****

Referee #1 (Remarks):

This manuscript is well revised for our concerns.

Referee #2 (Remarks):

This work has been greatly improved by the additional results following the careful revision. However, the length of the result section is too long, especially Figure 1, which should be more concise to fit the pattern of a report. And some conclusive and connective sentences displaying the logical relationship of each part should be added.

Corresponding Author Name:
Journal Submitted to:
Manuscript Number: